# Perioperative haemodynamic therapy for major gastrointestinal surgery: the effect of a Bayesian approach to interpreting the findings of a randomised controlled trial

Elizabeth G Ryan,[1,2] Ewen M Harrison,[3] Rupert M Pearse,[4] Simon Gates[1,2]

EGR and EMH contributed equally.

¹Warwick Clinical Trials Unit, University of Warwick, Coventry, UK
²Cancer Research UK Clinical Trials Unit, University of Birmingham, Birmingham, UK
³Clinical Surgery, University of Edinburgh, Edinburgh, UK
⁴Barts and the London School of Medicine & Dentistry, Queen Mary University of London, London, UK

**Correspondence to**
Dr Elizabeth G Ryan;
E.G.Ryan@bham.ac.uk

## ABSTRACT

**Objective** The traditional approach of null hypothesis testing dominates the design and analysis of randomised controlled trials. This study aimed to demonstrate how a simple Bayesian analysis could have been used to analyse the Optimisation of Perioperative Cardiovascular Management to Improve Surgical Outcome (OPTIMISE) trial to obtain more clinically interpretable results.

**Design, setting, participants and interventions** The OPTIMISE trial was a pragmatic, multicentre, observer-blinded, randomised controlled trial of 734 high-risk patients undergoing major gastrointestinal surgery in 17 acute care hospitals in the UK. Patients were randomly allocated to a cardiac output-guided haemodynamic therapy algorithm for intravenous fluid and inotropic drug administration during and in the 6 hours following surgery (n=368) or to standard care (n=366). The primary outcome was a binary outcome consisting of a composite of predefined 30-day moderate or major complications and mortality.

**Methods** We repeated the primary outcome analysis of the OPTIMISE trial using Bayesian statistical methods to calculate the probability that the intervention was superior, and the probability that a clinically relevant difference existed. We explored the impact of a flat prior and an evidence-based prior on our analyses.

**Results** Although OPTIMISE was not powered to detect a statistically significant difference between the treatment arms for the observed effect size (relative risk=0.84, 95% CI 0.70 to 1.01; p=0.07), by using Bayesian analyses we were able to demonstrate that there was a 96.9% (flat prior) to 99.5% (evidence-based prior) probability that the intervention was superior to the control.

**Conclusions** The use of a Bayesian analytical approach provided a different interpretation of the findings of the OPTIMISE trial (compared with the original frequentist analysis), and suggested patient benefit from the intervention. Incorporation of information from previous studies provided further evidence of a benefit from the intervention. Bayesian analyses can produce results that are more easily interpretable and relevant to clinicians and policy-makers.

**Trial registration number** ISRCTN04386758; Post-results.

## INTRODUCTION

The traditional statistical approach of null hypothesis testing dominates clinical trial design and analysis. In this *frequentist* framework, conclusions are drawn using p values and CIs which have been generated to test a specific hypothesis (usually a null hypothesis of exactly zero treatment difference). It requires that a correct conclusion be drawn with a high level of probability (statistical power) from a notional set of repetitions of the trial.

The problems with this approach are well described.[1–4] Frequentist results (p values and CIs) do not have simple or intuitive interpretations, and have often been misused and misinterpreted in clinical trials.[2] Randomised controlled trial results are often divided into 'significant' and 'non-significant', usually based on a p value <0.05. A statistically non-significant result for the primary endpoint is generally interpreted as the new treatment being 'unsuccessful'.[5 6] This is often an incorrect interpretation, and was highlighted by Altman and Bland,[7] 'absence of evidence is

not evidence of absence'. Randomised controlled trials are regarded as the gold standard of evidence and their results often inform clinical guidelines and practice. The misuse and misinterpretation of p values (and CIs) may result in the abandonment of potentially beneficial treatments. McShane *et al*[4] recommended that null hypothesis testing should be abandoned altogether, and one peer-reviewed journal has banned its use.[8]

Bayesian statistical methods provide an alternative framework for statistical modelling, and are increasingly being used in randomised controlled trials.[9–11] The Bayesian approach can be used at the design stage and for data analysis. Bayesian statistics provide a formal method for combining pre-existing information with data that are collected in the clinical trial into the analysis so that the current state of knowledge can be updated. This is particularly useful for clinical research given that advances in healthcare usually occur through incremental gains in knowledge. The Bayesian approach may also provide results that are easier to interpret than the frequentist approach since it computes the probability of various values of the treatment effect, given the data.

Pre-existing information is incorporated into the analysis via a *prior distribution*. This is a probability distribution that accounts for uncertainty in an unknown parameter, for example, the treatment effect, before the data from the clinical trial has been incorporated. The frequentist approach assumes the parameter to have a fixed, but unknown value. Some are put off by this aspect of the Bayesian approach, expressing a concern that it is not scientifically objective. However, any statistical method, including traditional frequentist methods, involve subjective choices. For instance, the arbitrarily set 5% significance level in the frequentist approach. Moreover, so-called *non-informative priors* can be used when there is little reliable previous information, or when one would prefer to numerically mimic a frequentist analysis and avoid introducing external information into the analysis. These are often used as a default prior.

When clinical trial data are observed, the prior distribution is updated to become the *posterior distribution*, which summarises the current knowledge. The posterior distribution of the treatment effect provides the relative credibility of the range of treatment effect values, given the trial data. Calculation of the posterior distribution can often be straightforward; however, many practical examples require computationally intensive approaches. Part of the reason Bayesian statistics is becoming more popular is due to the advances in computing power and computational techniques. The posterior distribution can be used to answer questions of direct relevance to decision-making, such as 'What is the probability that the new treatment is more successful than the current treatment?' Frequentist methods are limited as to what research questions they can answer, and cannot directly provide answers to clinically important questions.

Rather than using CIs, Bayesian statistics can use credible intervals, which provide a range of values for the treatment effect for a certain level of posterior probability. The highest posterior density interval (HDI) is the narrowest type of credible interval available for a specified probability. Although frequentist CI and Bayesian HDI may sometimes be numerically similar, their interpretation is different. A 95% HDI says, given the observed data, there is a 95% probability that the true parameter value falls within this interval. The frequentist interpretation of a 95% CI says, if this trial were to be repeated many times and CIs were calculated, 95% of these intervals would contain the true parameter value.

In this article we provide an example of how a simple Bayesian analysis of a binary primary outcome could have been used instead of (or in addition to) traditional frequentist methods, to give a better insight into the clinical application of the results. We analyse a randomised controlled trial of two treatments for patients undergoing major gastrointestinal surgery, to arrive at a conclusion that may be more clinically interpretable and useful, and also provide a more positive message from the trial.

## METHODS
### Data and study design
The Optimisation of Peri-operative Cardiovascular Management to Improve Surgical Outcome (OPTIMISE) trial[12] was a pragmatic, observer-blinded, multi-centre, randomised controlled trial of 734 high-risk patients (aged 50 years and older) that had undergone major gastrointestinal surgery in 17 acute care hospitals in the UK. Previous research suggested that postoperative outcomes may be improved by the use of cardiac output monitoring to guide the administration of intravenous fluid and inotropic drugs in the time around surgery.[13] Patients were randomly assigned (1:1) to a cardiac output-guided haemodynamic therapy algorithm for intravenous fluid and inotropic drug administration during and in the 6 hours following surgery (intervention; n=368) or to usual care (n=366). The primary outcome was a composite of predefined 30-day moderate or major complications and mortality.

The primary outcome occurred in 158/364 (43.41%) usual care and 134/366 (36.61%) intervention patients. At first glance, this would suggest that the results favour the intervention group compared with usual care. Using the traditional frequentist approach, the relative risk (RR) was found to be 0.84 (95% CI 0.7 to 1.01) and absolute risk reduction was 6.79% (95% CI −0.30% to 13.89%), with p=0.07. This was interpreted as there being insufficient evidence that the RR for complication/death at 30 days is different to 1. Thus, the authors concluded, "Use of this cardiac output-guided, hemodynamic therapy algorithm was not associated with a significant reduction in the composite primary outcome of moderate or major postoperative complications at 30 days following surgery."[12] A difference exists between the groups, but this was not judged to be sufficient using the conventional approach.

## Patient involvement

Patients were not involved in the re-analysis of the OPTI-MISE trial.

## Statistical methods

By repeating the same analysis using Bayesian methods we can find an alternative way to model and interpret these data. The Bayesian analysis involves specifying a prior distribution for the primary outcome rate in each treatment arm, and using the OPTIMISE trial data to update these prior distributions to become posterior distributions. From the posterior distributions we calculate the RR or absolute risk difference and 95% HDI. We also use the posterior distribution to calculate the probability that the intervention is superior (RR <1 or absolute risk difference <0).

We can extend this idea by specifying a clinically significant effect size. For instance, a group of clinicians or patients could decide that a RR must differ from the null by, say, 10% to be clinically significant. This 'region of practical equivalence' (ROPE) can be used to make probability statements about the likely clinical significance of effect sizes. For illustrative purposes we specify the ROPEs to be 0.9–1.1 for the RR and −0.05 to 0.05 (−5% to 5%) for the absolute risk difference. These ROPEs may not be reasonable in practice, and should be developed based on clinician, patient and health economists' feedback.

Additionally, we can simulate the outcomes of future patients using the updated estimates of the probabilities of having the primary outcome (from the posterior distribution) to predict the proportion of cases in which using the intervention would result in no complication/death while not using the intervention would.

We used two different types of prior distributions in separate analyses to check the robustness of our conclusions to our prior distribution assumptions. We used Beta distributions for the priors as these allowed the posterior distribution to be calculated more easily for our data. The sum of the two shape parameters in the Beta distribution provides an estimate of the effective sample size that the prior distribution provides, that is, how much information it contributes.

Initially we used a flat prior for both arms, the beta(1,1) distribution. This assumes that the prior distributions for the proportions for the composite outcome are uniform in each arm (see figure 1, top panel), that is, all values between 0 and 1 are equally likely *a priori*. While some values of the composite outcome rate are more likely than others, a flat prior may be used so that inferences from the posterior are driven by the trial data and probabilistic statements about the treatment effect can still be made. This prior contributes one additional patient with no complications and one patient with a complication/death. The results from this prior are unlikely to give different results to the original analysis, in terms of the RR and absolute risk difference, but the interpretations are likely to differ. A weakness of this approach is that too much prior probability is placed on extremely

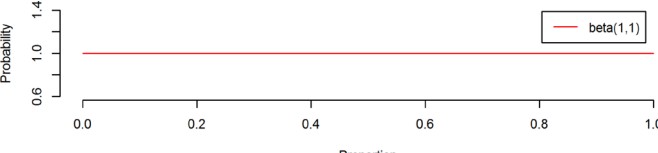

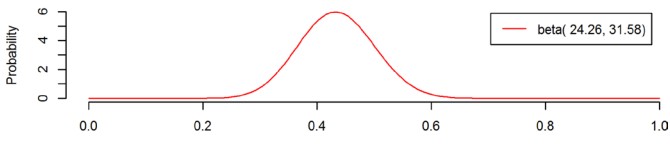

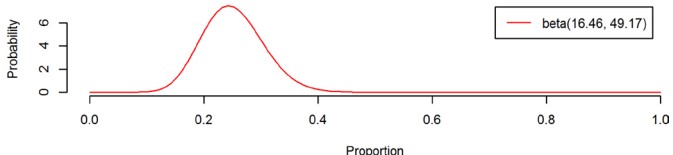

**Figure 1** Prior distributions used in Bayesian re-analysis of Optimisation of Perioperative Cardiovascular Management to Improve Surgical Outcome. The top panel is a flat prior distribution, beta(1, 1), that was used for both treatment arms. The middle and bottom panels show the evidence-based priors that were derived using a meta-analysis. The middle panel is the prior for the control arm (beta(24.26, 31.58)) and the bottom panel is the prior for the intervention arm (beta(16.46, 49.17)).

unlikely outcomes, that is, the prior described gives an equal weighting to the probability of all patients having a complication and no patients having a complication.

When the results from the OPTIMISE trial were incorporated into an updated meta-analysis, it was found that the intervention reduced the incidence of 30-day complications/death following surgery: 31.52% (intervention) versus 41.60% (control), RR 0.77 (95% CI 0.71 to 0.83).[12] We wanted to combine the previous information with the OPTIMISE trial data so that more precise treatment effect estimates could potentially be obtained and to pull the data away from inappropriate inferences. Therefore, an evidence-based prior was also specified using the results from a pre-existing systematic review[12 13] which had information on both treatment arms (online supplementary figures S1 and S2). Figure 1 shows the evidence-based prior distributions for the control (middle panel) and intervention arm (bottom panel), where the distribution for the primary outcome rate in the intervention arm is centred at a lower value. Further details on how these priors were derived are displayed in the online supplementary material. The evidence-based prior for the control arm contributed an effective sample size of 56 patients, and the intervention arm prior contributed 66 patients worth of information. With this additional information, the RR and absolute risk difference may decrease in favour of the intervention, compared with the original analyses.

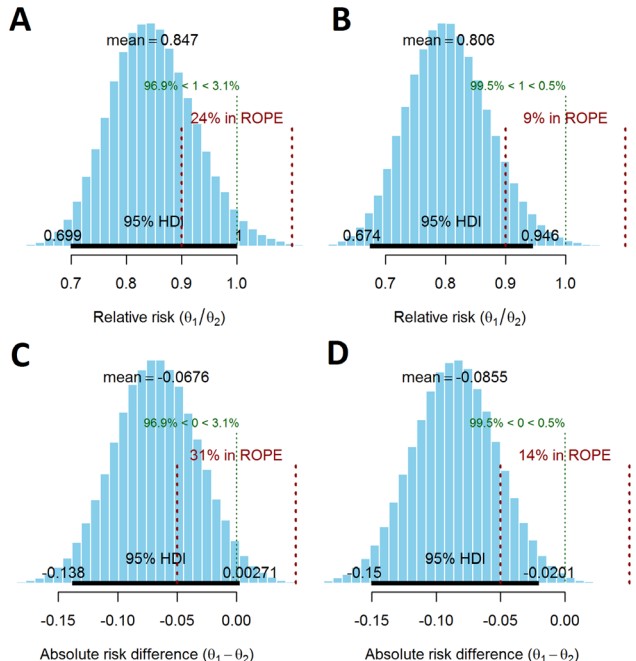

**Figure 2** Posterior distributions of relative risk (RR) and absolute risk difference for the flat prior and evidence-based priors. $\theta_1$ and $\theta_2$ are the composite outcome rates in the intervention and control arms, respectively. Posterior distribution of the RR of the primary outcome using (A) a flat prior, (B) evidence-based priors. A RR >1 indicates that the intervention is more harmful. Posterior distribution of the absolute risk difference in the primary outcome using (C) a flat prior (D) evidence-based priors. A positive value of the absolute risk difference indicates that the intervention is more harmful. The 95% highest posterior density interval (HDI) is shown as the thick black horizontal line with the boundaries written above the line. The region of practical equivalence (ROPE) is shown in red (with dotted vertical lines). The posterior probability of RR <1 and RR >1, and an absolute difference <0 and >0 is shown in green above the ROPE.

The Bayesian analysis was conducted in R V.3.4.1 and additional information on the algorithm settings and the code for the Bayesian analysis is available in the online supplementary material. Readers can perform the analysis here: https://argoshare.is.ed.ac.uk/bayesian_two_proportions/

## RESULTS

The results differ slightly between the flat and evidence-based priors, particularly for the probability of lying in the ROPE. Under the flat prior, the posterior mean RR is 0.85 (95% HDI 0.70 to 1.00), which is approximately the same as the frequentist analysis. Using the evidence-based priors, the posterior mean RR is 0.81 (95% HDI 0.67 to 0.95). The probability that the intervention group has a lower incidence of the composite endpoint is 96.9% and 99.5%, assuming a flat and evidence-based prior, respectively (figure 2A,B).

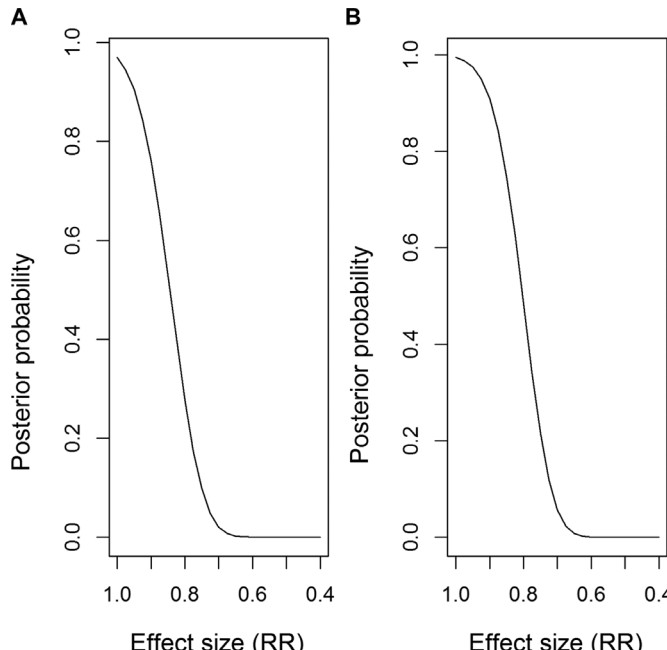

**Figure 3** Posterior probability of specified effect sizes, using (A) flat prior, (B) evidence-based priors. The line shows the probability of the relative risk (RR) being lower than the values on the x-axis (ie, a bigger treatment effect). A RR <1 indicates that the primary outcome rate is smaller in the intervention arm compared with the control arm.

Based on the RR ROPE, the probability of the two arms being clinically equivalent was 24% and 9% under the flat and evidence-based priors, respectively. Alternatively, the probability of there being a clinically relevant difference in the RR of the composite endpoint between the two groups was 76% and 91% for the flat and evidence-based priors, respectively.

Using the flat prior, the posterior mean of the absolute risk difference/reduction was 6.76% (95% HDI −0.30% to 13.80%), which is approximately the same as the frequentist analysis. Using the evidence-based prior, this value was 8.55% (95% HDI 2.00% to 15.00%). When the ROPE was defined as an absolute risk difference of −5% to 5%, the probabilities of the trial arms being equivalent were 31% and 14% (69% and 86% probability of there being a clinically relevant difference) using the flat and evidence-based priors, respectively (see figure 2C,D).

The posterior probability of different RRs occurring is shown in figure 3A (flat prior) and figure 3B (evidence-based prior). For example, in figure 3A, the probability that the RR <1 is 0.97, and the probability RR <0.8 is 0.28.

Using the Bayesian model, predictions were obtained from the posterior distribution for future patients. These found that the probability that a patient in the intervention group did not have a complication/death when a patient in the control group did have a complication/death was 27.56% (flat prior) and 28.49% (evidence-based prior). (The probability of a patient in the intervention group having a complication/death when a patient in the

control group did not was 20.77% and 19.59%, assuming a flat and evidence-based prior, respectively.)

## DISCUSSION

In this work we have performed a Bayesian re-analysis of the OPTIMISE trial[12] to demonstrate how a Bayesian approach can more easily answer clinical questions of interest and inform decision making. Our conclusions from the analysis are given in terms of probabilities that a benefit exists. The Bayesian approach also enabled us to formally incorporate information from previous studies on the event rates for each arm and combine this information with the OPTIMISE trial data. The principal finding of this analysis was that the use of a Bayesian analytical approach provided a different interpretation of the findings of the OPTIMISE trial, suggesting patient benefit from the intervention.

The OPTIMISE trial was powered (at 90%) to detect a reduction in the primary outcome rate from 50% in the control group to 37.5% in the intervention group (RR 0.75, absolute risk reduction 12.5%). The frequentist analysis conducted by Pearse *et al*[12] identified no significant difference for 30-day complications/death between the treatment arms and deemed the trial to be underpowered. The Bayesian posterior mean estimates and 95% HDI based on the flat priors gave similar results to the frequentist analysis as expected, with more optimistic results produced by the evidence-based prior. (Previous evidence had suggested that the intervention was superior.) The primary advantage here of using the Bayesian approach is that we can explicitly obtain the posterior distribution of the primary outcome rates for each arm, and from this the treatment effect distribution, which can be interpreted directly and intuitively. From these distributions we can calculate the probability that the new treatment is superior. A frequentist analysis, particularly of an underpowered trial, provides little help for clinicians to make the necessary decisions regarding future treatment and also runs the risk of concluding no difference between treatments (based on a non-significant p value), even when there is in reality a useful difference. By incorporating previous information into the prior distribution, the Bayesian approach can provide more informative probabilistic statements about the treatment effect.

For a reasonably large trial, different priors should have little effect on the posterior distributions as the data are contributing a large amount of information, and generally outweigh the contribution of the prior information. In this study, altering the priors had little impact on the results, most likely due to the fairly large sample size. The evidence-based prior used information from 21 small studies whose sample sizes ranged from 34 to 390 patients and were mostly in favour of the intervention. The evidence-based prior for the control arm contributed 56 patients worth of information compared with the 364 patients from OPTIMISE; the evidence-based prior for the intervention arm contributed 66 patients worth

of information and the OPTIMISE trial contributed 366 patients. The analyses which used the evidence-based priors obtained a slightly smaller RR and provided more certainty that the intervention was superior, compared with the flat priors. Since the previous studies were small studies, one may wish to down weight the influence of these studies in the prior to consider the risk of bias. Plots of the priors overlaid with the posteriors for the evidence-based priors are provided in the supplementary material (online supplementary figure S3) and show that the prior for the intervention was too optimistic. This is likely due to the small size of many of the previous studies and the large degree of heterogeneity between studies. Publication bias may also be present.

The Bayesian analyses that we have presented here are simple, and repeat the primary analysis of the OPTIMISE trial to demonstrate the usefulness of the Bayesian approach. These analyses could be extended to allow for more complicated models, such as Bayesian hierarchical models that account for site-specific effects or to allow for covariate adjustment. A Bayesian analysis cannot salvage a badly designed or conducted trial. Instead, the Bayesian approach can be used to produce results that are more easily interpreted and avoids the dichotomisations that occur in significance testing.

## CONCLUSION

Using the results from a standard statistical analysis, the OPTIMISE trial authors concluded that the use of the intervention compared with usual care did not significantly reduce a composite outcome of 30-day complications and mortality.[12] Although the trial was not powered to detect a statistically significant difference between the treatment arms for the observed effect size, by using Bayesian analyses we were able to demonstrate that the intervention was superior to the control with 96.9%–99.5% probability. These results are much more clinically useful than those that are given by the traditional approach.

**Contributors** EMH and EGR wrote the statistical code and performed the Bayesian analysis, interpreted the findings and wrote the manuscript. SG contributed to the writing of the manuscript. RMP was the CI for the OPTIMISE study, was the lead author on the primary OPTIMISE paper and contributed to the writing of the manuscript.

**Funding** EGR and SG were supported by a Medical Research Council (MRC) Methodology Research Grant (Grant number: MR/N028287/1). SG is supported by a National institute of Health Research Senior Investigator Award (NF-SI-0616-10008).

**Competing interests** RMP holds research grants, and has given lectures and/or performed consultancy work for Nestle Health Sciences, BBraun, Medtronic, Glaxo Smithkline, Intersurgical, and Edwards Lifesciences, and is a member of the Associate editorial board of the British Journal of Anaesthesia.

**Patient consent for publication** Not required.

**Ethics approval** OPTIMISE was approved by the East London and City Research Ethics Committee and the Medical and Healthcare Products Regulatory Agency.

**Provenance and peer review** Not commissioned; externally peer reviewed.

**Data sharing statement** Statistical code is available in the Supplementary Material.

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
