## [Reviewer comments · BMJ Open]

ARTICLE DETAILS

TITLE (PROVISIONAL)	Perioperative haemodynamic therapy for major gastrointestinal surgery: The effect of a Bayesian approach to interpreting the findings of a randomised controlled trial
AUTHORS	Ryan, Elizabeth; Harrison, Ewen; Pearse, Rupert; Gates, Simon

VERSION 1 - REVIEW

REVIEWER	Alexina Mason London School of Hygiene and Tropical Medicine
REVIEW RETURNED	13-Aug-2018

GENERAL COMMENTS	This paper describes a re-analysis of the primary outcome of the OPTIMISE trial using Bayesian methods. The paper is well written and makes some important points about the ability of a Bayesian approach to provide direct answers to clinically important questions and present trial results in more easily interpretable ways compared to a traditional frequentist approach. The authors present results from two priors, one “non-informative” (but see below) and the other informative, but the motivation underlying the priors is not clear and needs more discussion. Further discussion about the differences between the two priors, and their appropriateness would enhance the paper. Why was the evidence-based prior for the intervention “too optimistic”? How similar were the trials in the meta-analysis to OPTIMISE? Event rates often change over time and meta-analyses may suffer from publication bias – how could these be taken into account in choice of prior, or by a sensitivity analysis? It would also be helpful to discuss how the two analyses are expected to differ from the original frequentist analysis, and the relative merits of incorporating information from previous trials through an informative prior versus carrying out an updated meta-analysis, which I note is how the original OPTIMISE paper summarised the state of knowledge post their report. Paragraph 1 of the discussion could be extended to point out that the Bayesian approach enabled formal use of available information about the event rates in each arm. Related to this, Paragraph 3 of the discussion could be strengthened by including a discussion about the amount of information that each prior contributes to the analysis. (For a Beta-binomial model, as used in this analysis, the sum of the two Beta parameters can be interpreted as an ‘effective sample size’, e.g. for the evidence-based prior for the control arm 56 patients worth of data is being incorporated through the prior compared with 364 patients from OPTIMISE. See page 37 of the BUGS book for further detail.) From the abstract it should be clear that different results are being obtained because one of the priors brings extra information into
--

	the analysis. Just by switching to a Bayesian analysis will not gain power. The Bayesian approach allowed the incorporation of other relevant information as well as providing a 'different interpretation'. The abstract conclusion should reflect this. The terms 'non-informative prior' (e.g. p3, line 44) and 'uninformative prior' (e.g. p5, line 30) should be avoided as they can be misleading. I suggest referring to your Beta(1,1) prior as a flat prior on the probability scale – it provides the information that all values between 0 and 1 are considered equally likely a priori. Other comments p3, line 38: a prior distribution is described as a “probability distribution that accounts for uncertainty in an unknown parameter” – to distinguish it from a posterior distribution add 'excluding the evidence from the clinical trial' or similar. p3, line 39: for completeness, “frequentist approach assumes the parameter to have a fixed value” is better worded 'fixed, but unknown value'. p3, line 45: when discussing mimicking a frequentist analysis, be clear this is numerically – a Bayesian would interpret the numbers very differently to a frequentist. p4, line 2: the description of a HDI is for a more general posterior interval, often referred to as a credible interval. A HDI is the narrowest credible interval available with specified probability. p4, line 45: the authors' conclusions from the original paper are a clear misinterpretation of a frequentist analysis – precisely the point made in para 2 of the introduction and worth linking. To be fair to the original authors, a more correct frequentist interpretation is given in their discussion section. p5, line 30: “which assumes that the proportions for the composite outcome are uniform in each group” is more accurately “which assumes that the prior distributions for the proportions ...” Figure 3: I find this hard to interpret – more explanation in the text and title would help. p6, last para of results: this is quite hard to think through, and an explanation of posterior prediction might help a non-statistical audience. For balance, what about the flip side? (Probability that a patient in the control group did not have a complication/death when a patient in the intervention group did.) p6, discussion, paragraph 2: what effect size was OPTIMISE powered to find? Conclusion: this should recognise that the OPTIMISE trial authors also carried out a meta-analysis which led to a different conclusion from the one based on the OPTIMISE data alone. As an aside, there is considerable heterogeneity amongst the studies included in the meta-analysis; the implications of this and the possibility of publication bias for prior choice deserve a mention. Use a consistent number of decimal places throughout (e.g. p5, line 56; p6, line 4 and p6, line 14). Reference: Lunn D, Jackson C, Best N, Thomas A, Spiegelhalter D. The BUGS book: A practical introduction to Bayesian analysis. Chapman & Hall. 2013.
--	--

REVIEWER	Haolun Shi University of Hong Kong, Hong Kong
REVIEW RETURNED	17-Aug-2018

GENERAL COMMENTS	The paper provides an alternative way to analyze the results of the OPTIMISE trial to obtain more clinically interpretable results. The authors take a Bayesian approach in the interpretation of the trial. Overall, the paper is wellmotivated and the presentation is clear. The Bayesian methodology and the authors' interpretation are sound and justifiable. The authors may consider improving the paper with respect to the following aspects.  1. It would be better if the type of the primary endpoint is clearly indicated in the abstract, i.e., binary. 2. It would be better if the author can provided a brief introduction on the motivation behind using the Bayesian approach instead of the frequentist one for the OPTIMISE trial. 3. It seems that evidence-based prior is too optimistic. The authors may consider commenting on the possible difference in trial conducts and patients' prognostic factors between the current trial and the historical trials. More justification of using such an evidence-based prior is needed. 4. It seems that using JAGS for uninformative Beta prior is unnecessary. The posterior distribution is also Beta. Hence there is no need to do MCMC chaining. 5. As this is a multi-center trial, the author may consider discussing the possibility of using Bayesian hierarchical model. 6. The last sentence in the Discussion section "A frequentist analysis, particularly of an underpowered trial, provides little help for clinicians to make the necessary decisions...", I believe for an underpowered trial, a Bayesian approach may also suffer from the same problem, unless an optimistic prior is adopted.
--

VERSION 1 – AUTHOR RESPONSE

Response to Reviewers

We are grateful to the reviewers and the assistant editor for a thorough review of this manuscript and for their helpful feedback and suggestions. Below is a point-by-point response to all major and minor comments and a description of how the manuscript has been revised.

Reviewer 1

Comments to authors

This paper describes a re-analysis of the primary outcome of the OPTIMISE trial using Bayesian methods. The paper is well written and makes some important points about the ability of a Bayesian approach to provide direct answers to clinically important questions and present trial results in more easily interpretable ways compared to a traditional frequentist approach.

The authors present results from two priors, one "non-informative" (but see below) and the other informative, but the motivation underlying the priors is not clear and needs more discussion.

Further discussion about the differences between the two priors, and their appropriateness would enhance the paper.

We have added more detail to the paragraphs on the prior distributions in the Statistical Methods section (p6-7), describing the purpose of each prior distribution, why they might be used and the amount of information they contribute:

“We used two different types of prior distributions in separate analyses to check the robustness of our conclusions to our prior distribution assumptions. We used Beta distributions for the priors as these allowed the posterior distribution to be calculated more easily for our data. The sum of the two shape parameters in the Beta distribution provides an estimate of the effective sample size that the prior distribution provides, i.e. how much information it contributes.

Initially we used a flat prior for both arms, the Beta(1,1) distribution. This assumes that the prior distributions for the proportions for the composite outcome are uniform in each group (see Figure 1, top panel), that is, all values between 0 and 1 are equally likely a priori. Whilst some values of the composite outcome rate are more likely than others, a flat prior may be used so that inferences from the posterior are driven by the trial data and probabilistic statements about the treatment effect can still be made. This prior contributes one additional patient with no complications and one patient with a complication/death. The results from this prior are unlikely to give different results to the original analysis, in terms of the RR and absolute risk difference, but the interpretations are likely to differ. A weakness of this approach is that too much prior probability is placed on extremely unlikely outcomes, i.e., the prior described gives an equal weighting to the probability of all patients having a complication and no patients having a complication.

When the results from the OPTIMISE trial were incorporated into an updated meta-analysis, it was found that the intervention reduced the incidence of 30-day complications/death following surgery: 31.52% (intervention) vs 41.60% (control), RR 0.77 (95% CI: 0.71, 0.83) [12]. We wanted to combine the previous information with the OPTIMISE trial data so that more precise treatment effect estimates could potentially be obtained and to pull the data away from inappropriate inferences. Therefore, an evidence-based prior was also specified using the results from a pre-existing systematic review ([12, 13]) which had information on both treatment arms. Figure 1 shows the evidence-based prior distributions for the control (middle panel) and intervention arm (bottom panel), where the distribution for the primary outcome rate in the intervention arm is centred at a lower value. Further details on how these priors were derived are displayed in the Supplementary Material. The evidence-based prior for the control arm contributed an effective sample size of 56 patients, and the intervention arm prior contributed 66 patients worth of information. With this additional information, the RR and absolute risk difference may decrease in favour of the intervention, compared to the original analyses.”

We also altered the final sentence of paragraph 4 in the Introduction (p4) to read:

“Moreover, so-called non-informative priors can be used when there is little reliable previous information, or when one would prefer to numerically mimic a frequentist analysis and avoid introducing external information into the analysis. These are often used as a default prior.”

Why was the evidence-based prior for the intervention “too optimistic”?

Many of the previous studies were small and there was a large degree of heterogeneity between the studies. It is likely that publication bias was also present. The evidence-based priors have incorporated uncertainty and are not simply the distribution of the previous data.

How similar were the trials in the meta-analysis to OPTIMISE? Event rates often change over time and meta-analyses may suffer from publication bias – how could these be taken into account in choice of prior, or by a sensitivity analysis?

The studies included in the meta-analysis were heterogeneous, and smaller than OPTIMISE. The most competent trials in the systematic review were small with a high risk of bias, and were viewed as “hypothesis generating” rather than confirmatory. We chose not to perform any sensitivity analyses given the small contribution of each study to the overall effect estimate.

There is no clear best approach of how to include historical information into the design and analysis of clinical trials. The approach that we used to derive our priors (described in the supplementary

material) was to perform a Bayesian meta-analysis on the pre-existing studies, and account for the heterogeneity between the trials. This information was down-weighted since we did not simply use the overall event rates for each arm (456/1111 for control and 354/1180 for intervention) in the prior distributions.

One could further down-weight this information by instead using a “weakly informative” prior and use the estimates from these studies as a guide as to where the centre of the distribution for the primary outcome rates is likely to be, but may wish to increase the variance of the distribution.

Alternatively, power priors could be used (e.g., Chen et al., 2000, *Stat Plan Infer*; Neuenschwander et al., 2009, *Statistics in Medicine*), which incorporate a weight parameter, which may be fixed or unknown, that determines how much information is used from the historical data.

We have edited the “Evidence-based Prior Section” in the Supplementary materials to include this:

“An evidence-based prior was specified using results from a pre-existing meta-analysis ([4,5] – not including the OPTIMISE results) which had information on both treatment arms. There is no clear best approach of how to include historical information into the analysis of a clinical trial. Viele et al. [6] provide a good overview of the available approaches when data for the control arm is available and discuss how to decide to what extent historical data should be incorporated.

The R Bayesian evidence synthesis Tools (RBeST; [7]) were used to perform a Bayesian meta-analysis, which accounts for the uncertainty of the population mean and between-trial heterogeneity (using the gMAP function in R (version 3.4.1)). We used this approach as it is a principled and reproducible method of combining data from previous trials. Further refinements could also be included, such as attempting to correct for publication or other biases....”

We have also added the following to the end of the “Evidence-based Prior Section” in the Supplementary Material:

“We note that power priors [9] could have been used instead for deriving the evidence-based priors, which incorporate a weight parameter that determines how much information is used from the historical data.”

It would also be helpful to discuss how the two analyses are expected to differ from the original frequentist analysis, and the relative merits of incorporating information from previous trials through an informative prior versus carrying out an updated meta-analysis, which I note is how the original OPTIMISE paper summarised the state of knowledge post their report.

We have added the following to the end of the paragraph discussing the flat prior distribution (p6):

“...The results from this prior are unlikely to give different results to the original analysis, in terms of the RR and absolute risk difference, but the interpretations are likely to differ. A weakness of this approach is that too much prior probability is placed on extremely unlikely outcomes, i.e., the prior described gives an equal weighting to the probability of all patients having a complication and no patients having a complication.”

We have also added the following to the paragraph describing the evidence-based priors (top of p7):

“Figure 1 shows the evidence-based prior distributions for the control (middle panel) and intervention arm (bottom panel), where the distribution for the primary outcome rate in the intervention arm is centred at a lower value.... The evidence-based prior for the control arm contributed an effective sample size of 56 patients, and the intervention arm prior contributed 66 patients worth of information. With this additional information, the RR and absolute risk difference may decrease in favour of the intervention, compared to the original analyses.”

The focus of this paper was not the meta-analysis, but rather how a Bayesian approach to analysing RCTs can lead to a different interpretation of the trial results. One of the major advantages to incorporating historical information in the prior distribution is that the prior can be used from the start of the trial (or even in the design phase) and so the historical information can contribute to any decisions that are made as the trial progresses (e.g., in interim analyses). This is a more efficient use of the information. An updated meta-analysis could not be performed until the trial was completed. The relative contribution/weighting of each study to the overall estimate may differ between the two approaches, and it's likely that the previous studies would have less weight in a Bayesian analysis of a trial compared to an updated meta-analysis.

Paragraph 1 of the discussion could be extended to point out that the Bayesian approach enabled formal use of available information about the event rates in each arm. Related to this, Paragraph 3 of the discussion could be strengthened by including a discussion about the amount of information that each prior contributes to the analysis. (For a Beta-binomial model, as used in this analysis, the sum of the two Beta parameters can be interpreted as an 'effective sample size', e.g. for the evidence-based prior for the control arm 56 patients worth of data is being incorporated through the prior compared with 364 patients from OPTIMISE. See page 37 of the BUGS book for further detail.)

We have added the following to paragraph 1 of the Discussion (p8):

"...The Bayesian approach also enabled us to formally incorporate information from previous studies on the event rates for each arm and combine this information with the OPTIMISE trial data."

We have provided more information about the ESS of the priors in paragraphs 5 and 6 of the Statistical Methods section (p6-7; see above).

We have altered the 3rd paragraph of the Discussion (p8) to:

"...The evidence-based prior used information from 21 small studies whose sample sizes ranged from 34 to 390 patients and were mostly in favour of the intervention. The evidence-based prior for the control arm contributed 56 patients worth of information compared to the 364 patients from OPTIMISE; the evidence-based prior for the intervention arm contributed 66 patients worth of information and the OPTIMISE trial contributed 366 patients."

From the abstract it should be clear that different results are being obtained because one of the priors brings extra information into the analysis. Just by switching to a Bayesian analysis will not gain power. The Bayesian approach allowed the incorporation of other relevant information as well as providing a 'different interpretation'. The abstract conclusion should reflect this.

We have added the following to the end of the "Methods" section in the Abstract:

"We explored the impact of a flat prior and an evidence-based prior on our analyses."

We have also added the following to the "Conclusion" section of the Abstract:

"Incorporation of information from previous studies provided further evidence of a benefit from the intervention."

Use of a flat prior still gave a 97% probability of patient benefit from the intervention.

The terms 'non-informative prior' (e.g. p3, line 44) and 'uninformative prior' (e.g. p5, line 30) should be avoided as they can be misleading. I suggest referring to your Beta(1,1) prior as a flat prior on the probability scale – it provides the information that all values between 0 and 1 are considered equally likely a priori.

We have changed these to "flat prior".

Other comments

p3, line 38: a prior distribution is described as a “probability distribution that accounts for uncertainty in an unknown parameter” – to distinguish it from a posterior distribution add ‘excluding the evidence from the clinical trial’ or similar.

We have added the following to the end of the abovementioned sentence:

“..., before the data from the clinical trial has been incorporated”.

p3, line 39: for completeness, “frequentist approach assumes the parameter to have a fixed value” is better worded ‘fixed, but unknown value’.

We have altered this sentence to:

“The frequentist approach assumes the parameter to have a fixed, but unknown value.”

p3, line 45: when discussing mimicking a frequentist analysis, be clear this is numerically – a Bayesian would interpret the numbers very differently to a frequentist.

We have changed this to:

“Moreover, so-called non-informative priors can be used when there is little reliable previous information, or when one would prefer to numerically mimic a frequentist analysis and avoid introducing external information into the analysis. These are often used as a default prior.”

p4, line 2: the description of a HDI is for a more general posterior interval, often referred to as a credible interval. A HDI is the narrowest credible interval available with specified probability.

We have altered the start of the 6th paragraph in the Introduction section (top of p5) to read: “Rather than using confidence intervals (CIs), Bayesian statistics can use credible intervals, which provide a range of values for the treatment effect for a certain level of posterior probability. The highest posterior density interval (HDI) is the narrowest type of credible interval available for a specified probability.”

p4, line 45: the authors’ conclusions from the original paper are a clear misinterpretation of a frequentist analysis – precisely the point made in para 2 of the introduction and worth linking. To be fair to the original authors, a more correct frequentist interpretation is given in their discussion section.

For simplicity, we have changed the quote in paragraph 2 of the Data and Study Design section to the statement that was given in the discussion in the original paper:

“...use of this cardiac output–guided, hemodynamic therapy algorithm was not associated with a significant reduction in the composite primary outcome of moderate or major postoperative complications at 30 days following surgery.”

p5, line 30: “which assumes that the proportions for the composite outcome are uniform in each group” is more accurately “which assumes that the prior distributions for the proportions ...”

We thank the reviewer for spotting this error and have incorporated their suggestion.

Figure 3: I find this hard to interpret – more explanation in the text and title would help.

We have altered the Figure 3 legend to read:

“Figure 3. Posterior probability of specified effect sizes, using (a) flat prior, (b) evidence-based priors. The line shows the probability of the RR being lower than the values on the x-axis (i.e., a bigger

treatment effect). A $RR < 1$ indicates that the primary outcome rate is smaller in the intervention arm compared to the control arm.”

We have also added the following to the end of the 1st paragraph in the results section:

“The posterior probability of different RRs occurring is shown in Figure 3 (for both priors). For example, in Figure 3(a), the probability that the $RR < 1$ is 0.97, and the probability $RR < 0.8$ is 0.28.”

p6, last para of results: this is quite hard to think through, and an explanation of posterior prediction might help a non-statistical audience. For balance, what about the flip side? (Probability that a patient in the control group did not have a complication/death when a patient in the intervention group did.)

We introduced the concept of posterior predictions in paragraph 3 of the Statistical Methods section. We have also altered the final paragraph of the results (p7-8) to the following:

“Using the Bayesian model, predictions were obtained from the posterior distribution for future patients. These found that the probability that a patient in the intervention group did not have a complication/death when a patient in the control group did have a complication/death was 27.40% (flat prior) and 28.26% (evidence-based prior). (The probability of a patient in the intervention group having a complication/death when a patient in the control group did not was 20.76% and 19.72%, assuming a flat and evidence-based prior, respectively).”

p6, discussion, paragraph 2: what effect size was OPTIMISE powered to find?

We have added the following to the beginning of paragraph 2 of the discussion (p8):

“The OPTIMISE trial was powered (at 90%) to detect a reduction in the primary outcome rate from 50% in the control group to 37.5% in the intervention group (RR 0.75, absolute risk reduction 12.5%).”

Conclusion: this should recognise that the OPTIMISE trial authors also carried out a meta-analysis which led to a different conclusion from the one based on the OPTIMISE data alone. As an aside, there is considerable heterogeneity amongst the studies included in the meta-analysis; the implications of this and the possibility of publication bias for prior choice deserve a mention.

The previous meta-analysis was not a major focus of this work, but simply provided information for an evidence-based/informative prior. The purpose of this work was to demonstrate how a Bayesian analysis can provide alternative and more clinically-relevant interpretations of the OPTIMISE trial results than the frequentist approach. The updated meta-analysis was not a focus of the original paper and was only added at the request of the journal editors.

We have instead mentioned the meta-analysis when describing the evidence-based prior in the Statistical Methods section (bottom of p6): “When the results from the OPTIMISE trial were incorporated into an updated meta-analysis, it was found that the intervention reduced the incidence of 30-day complications/death following surgery: 31.52% (intervention) vs 41.60% (control), RR 0.77 (95% CI: 0.71, 0.83) [12].”

We have added further detail on using historical information in priors to the Supplementary material (see above).

Use a consistent number of decimal places throughout (e.g. p5, line 56; p6, line 4 and p6, line 14).

We thank the reviewer for spotting this error. We have gone through the manuscript and edited the number of decimal places to make it more consistent.

Reference:

Lunn D, Jackson C, Best N, Thomas A, Spiegelhalter D. The BUGS book: A practical introduction to Bayesian analysis. Chapman & Hall. 2013.

Reviewer 2

The paper provides an alternative way to analyze the results of the OPTIMISE trial to obtain more clinically interpretable results. The authors take a Bayesian approach in the interpretation of the trial. Overall, the paper is well-motivated and the presentation is clear. The Bayesian methodology and the authors' interpretation are sound and justifiable. The authors may consider improving the paper with respect to the following

aspects.

1. It would be better if the type of the primary endpoint is clearly indicated in the abstract, i.e., binary.

The final sentence of the "Design, setting, participants, and interventions" section of the abstract has been altered to:

"The primary outcome was a binary outcome that consisted of a composite of predefined 30-day moderate or major complications and mortality."

2. It would be better if the author can provided a brief introduction on the motivation behind using the Bayesian approach instead of the frequentist one for the OPTIMISE trial.

The introduction section contains the motivation behind using the Bayesian approach for clinical trials and the issues that arise in using p-values, particularly with regards to their interpretation.

We have added the following to the second paragraph of the "Data and Study Design" section (underlined):

"The primary outcome occurred in 158/364 (43.41%) usual care and 134/366 (36.61%) intervention patients. At first glance, this would suggest that the results favour the intervention group compared with usual care. Using the traditional frequentist approach, the relative risk (RR) was found to be 0.84 (95% CI: 0.70, 1.01) and absolute risk reduction was 6.79% (95% CI: -0.30%, 13.89%), with $p=0.07$. This was interpreted as there being insufficient evidence that the RR for complication/death at 30 days is different to 1. Thus, the authors concluded, "use of this cardiac output-guided, hemodynamic therapy algorithm was not associated with a significant reduction in the composite primary outcome of moderate or major postoperative complications at 30 days following surgery." [12] A difference exists between the groups, but this was not judged to be sufficient using the conventional approach."

3. It seems that evidence-based prior is too optimistic. The authors may consider commenting on the possible difference in trial conducts and patients' prognostic factors between the current trial and the historical trials. More justification of using such an evidence-based prior is needed.

See above comments to Reviewer 1

4. It seems that using JAGS for uninformative Beta prior is unnecessary. The posterior distribution is also Beta. Hence there is no need to do MCMC chaining.

Yes, this example with the binary outcome (binomial likelihood) and conjugate prior does not require the MCMC algorithm to estimate the posterior distribution, which can be calculated using an analytic expression. We wanted to provide the MCMC algorithm code to help educate practitioners/clinicians

who may want to run MCMC algorithms for different models or using non-conjugate priors. Our web-app allows alternative priors and data to be specified and generates a posterior using a closed-form expression (https://argoshare.is.ed.ac.uk/bayesian_two_proportions/).

5. As this is a multi-center trial, the author may consider discussing the possibility of using Bayesian hierarchical model.

We have added the following to the last paragraph of the discussion:

“These analyses could be extended to allow for more complicated models, such as Bayesian hierarchical models that account for site-specific effects or to allow for covariate adjustment.”

6. The last sentence in the Discussion section “A frequentist analysis, particularly of an underpowered trial, provides little help for clinicians to make the necessary decisions...”, I believe for an underpowered trial, a Bayesian approach may also suffer from the same problem, unless an optimistic prior is adopted.

We have added the following to the end of the second paragraph in the Discussion section (p8):

“By incorporating previous information into the prior distribution, the Bayesian approach can provide more informative probabilistic statements about the treatment effect.”

A traditional analysis of an underpowered trial will not tell you much if you just look at non-significance, whereas a Bayesian analysis will quantify how much less uncertain you have become as a result of the “small” trial.

VERSION 2 – REVIEW

REVIEWER	Alexina Mason LSHTM, UK
REVIEW RETURNED	18-Oct-2018

GENERAL COMMENTS	I thank the authors for their detailed response to my previous comments. These have been carefully addressed in their manuscript, and I have only the following minor follow-up comments that they might like to consider. 1) p 5, line 24: revert to 0.9 to 1.1, instead of 0.90 to 1.10 2) p 6, line 34: replace 99.50% by 99.5% 3) p 7, line 29: add the words "based on the flat priors" or similar, so the first part of the sentence reads "The Bayesian posterior mean estimates and 95% HDI based on the flat priors gave similar results to the frequentist analysis as expected" 4) p 8, line 4 (last sentence of para 3 of discussion): the added words do not explain why the evidence-based prior for the intervention was too optimistic, merely restate what can be seen from the plot. I suggest replacing with the explanation you provided in your response to reviewers. 5) Conclusion: for consistency with abstract and results replace 97-99.5% with 96.9-99.5%
---

REVIEWER	Haolun Shi University of Hong Kong, Hong Kong
REVIEW RETURNED	07-Oct-2018

GENERAL COMMENTS	I recommend acceptance.
-------------------------

VERSION 2 – AUTHOR RESPONSE

Reviewer: 1

I thank the authors for their detailed response to my previous comments. These have been carefully addressed in their manuscript, and I have only the following minor follow-up comments that they might like to consider.

1) p 5, line 24: revert to 0.9 to 1.1, instead of 0.90 to 1.10

Thank-you for spotting this. We have made the changes you suggested.

2) p 6, line 34: replace 99.50% by 99.5%

Thank-you for spotting this. We have made your suggested correction.

3) p 7, line 29: add the words "based on the flat priors" or similar, so the first part of the sentence reads "The Bayesian posterior mean estimates and 95% HDI based on the flat priors gave similar results to the frequentist analysis as expected"

We thank the reviewer for this suggestion which we have now incorporated.

4) p 8, line 4 (last sentence of para 3 of discussion): the added words do not explain why the evidence-based prior for the intervention was too optimistic, merely restate what can be seen from the plot. I suggest replacing with the explanation you provided in your response to reviewers.

We thank the reviewer for this suggestion. We have deleted "...since it is centred at a lower primary outcome rate than the posterior" from the last part of this sentence and have instead added

"This is likely due to the small size of many of the previous studies and the large degree of heterogeneity between studies. Publication bias may also be present."

5) Conclusion: for consistency with abstract and results replace 97-99.5% with 96.9-99.5%

We thank the reviewer for spotting this and have changed this to 96.9-99.5%

Reviewer: 2

I recommend acceptance.

We thank the reviewer for reading our revised manuscript.